# Current Methods and Advances in the Immunotherapy Treatment of Non-Ovarian Gynaecological Cancers

Sola Adeleke [1,2,*], Yujia Gao [3], Somto Okoli [4], Sunyoung Choi [5], Hao Ding [6], Joao R. Galante [1] and Christos Mikropoulos [7]

1   Department of Oncology, Guy's and St Thomas' NHS Foundation Trust, London SE1 9RT, UK
2   School of Cancer & Pharmaceutical Sciences, King's College London, London WC2R 2LS, UK
3   UCL Medical School, University College London, London WC1E 6BT, UK
4   Bristol Medical School, University of Bristol, Bristol BS8 1QU, UK
5   GKT School of Medicine, King's College London, London WC2R 2LS, UK
6   School of Medicine, Imperial College London, London SW7 2DD, UK
7   Royal Surrey County Hospital, Guildford GU2 7XX, UK
*   Correspondence: olusola.adeleke@kcl.ac.uk

**Abstract:** Endometrial cancer (EC) and cervical cancer (CC) are common malignancies in women in clinical practice. More uncommon non-ovarian malignancies, such as vulval cancer (VC), are also becoming more prevalent in women of all ages. Currently, there are few comprehensive reviews on the management of these conditions, despite the recent advances in the use of immunotherapy in the management of other forms of cancer. The treatment modalities for EC, CC and VC vary; however, platinum-based chemotherapy, surgical resection and radiotherapy are the main forms of treatment. In more advanced or recurrent disease, there is a limited number of efficacious treatments, with many clinicians relying on adjuvant chemotherapy despite the increased rationale for the use of immunotherapy. With the development of the novel adoptive T-cell therapy, intra-tumoural oncolytic viral therapy and cancer vaccines, the landscape of gynaecological cancer management is changing, and it is likely that treatment efficacy and outcomes will improve dramatically. This review aims to summarise the current management of endometrial, cervical and vulval cancer and to evaluate the novel therapies under development, as well as the future of the management of non-ovarian gynaecological malignancies.

**Keywords:** endometrial cancer; cervical cancer; vulval cancer; immunotherapy; immune checkpoint inhibitors; chemotherapy

## 1. Introduction

Over recent years, immunotherapy has emerged as the mainstay of treatment in numerous oncological diseases. Through the use of immune checkpoint inhibitors, cancer vaccines and adoptive cell transfer, therapeutic effects can be established through upregulating or downregulating the immune system. Immunotherapy, specifically in the form of immune checkpoint inhibitors, unleashes the body's immune system against cancer and has been revolutionary in the treatment of several solid-tumour and haematological malignancies. These results have not been fully replicated in gynae-oncological conditions, and it is important to acknowledge the deficits in knowledge of their use, efficacy and safety compared to other oncological conditions [1]. However, a number of articles have explored the use of immunotherapy in the treatment of these conditions. Most comprehensive reviews published to date have focused mainly on ovarian cancer, with a few on endometrial and cervical cancers. Rarer gynaecological malignancies, such as vulval cancer, are often neglected, despite the advancements in recent years [2–4]. In this review, we will focus exclusively on non-ovarian gynaecological malignancies. This will provide the

opportunity to delve into more detail about various immunotherapy indications, clinical trial evidence, new therapeutic strategies currently in development and future directions.

Endometrial cancer (EC) is one of the most common gynaecological malignancies in developed countries, with both its incidence and mortality rising worldwide [5]. Women over the age of 50 account for more than 90% of EC cases, with the median age at diagnosis being 63 years. Nonetheless, 4% of EC patients are under the age of 40, with many of them still wishing to retain fertility [6]. The majority of EC cases are diagnosed at early stages, with a 5-year survival rate over 95%, whereas in advanced and metastatic diseases, which represent up to 10–15% of EC cases, the survival rates are much lower (68% if there is regional spread and 17% with distant metastases) [7,8]. Women with obesity, polycystic ovary syndrome (PCOS), diabetes mellitus, infertility, nulliparity or late menopause, or a history of unopposed oestrogen therapy and tamoxifen therapy are at an increased risk for EC, due to excess endogenous or exogenous oestrogen without adequate progestin opposition [7,9–12]. In 30% and 13–30% of primary and recurrent EC cases, respectively, the tumour is presented as MSI/dMMR (microsatellite instability/MMR-deficient), and the recognition of a tumour's MMR status is now of high clinical and prognostic significance [13].

In 2020, cervical cancer (CC) was the most prevalent gynaecological malignancy in women [14], ranking fourth globally in terms of both incidence and mortality [15,16]. Being the second most common non-dermatological cancer after breast cancer in women, it manifests more commonly in low- and middle-income countries (LMICs), mostly in sub-Saharan Africa, Melanesia and Southeast Asia [14,17]. Only 2% of cervical cancer cases present with distant metastatic disease, and the prognosis is poor, with a median survival of only 7 months [17]. The human papilloma virus (HPV) is well-known to be associated with cervical cancer [18]. For instance, HPV-16 and HPV-18 account for 71% of all newly diagnosed cases, whilst HPV types 31, 33, 45, 52 and 58 account for another 19% [16,19]. The two cancer types where this association is most pronounced are squamous cell carcinomas and cervical adenocarcinomas [16,19]. While HPV infection often emerges in sexually active young women between the ages of 18 and 30, cervical cancer is more likely in those over 35. This suggests that HPV infection occurs early in life and develops into cancer gradually [20]. The goal of elimination of cervical cancer ($\leq$4 per 100,000 women worldwide) has been proposed by the World Health Organization (WHO) in 2018 [21]. A triple-intervention strategy of HPV vaccination, screening programmes and timely treatment of precancerous lesions has been shown to be effective across several LMICs. It is anticipated to reduce the premature mortality rate of CC by one-third by 2030 and save the lives of more than 62 million women over the course of the next century [22,23]. Women are advised to start screening for CC from age 25 and, as a preferred option, to have primary HPV testing every three years until age 49 and every five years from 50 to 64 years old according to the National Institute for Health and Care Excellence (NICE) recommendations [24]. Studies have revealed that HPV-negativity (5.5–11% of CC) [25,26] has been on the decline [27], which could be attributed to advancements in HPV testing and non-cervical cancer classification [28]. Nonetheless, the absence of HPV infection was significantly associated with poor overall survival and prognosis [29,30], warranting additional research into the impact of negative HPV testing on prognosis and the development of appropriate treatments [28].

Vulval carcinoma (VC), an uncommon form of gynaecological cancer, has an incidence of 45,000 worldwide, accounting for 4% of gynaecological cancers and 0.3% of all newly diagnosed malignancies [15,31]. Like EC, VC typically develops in post-menopausal women, with a median age of 69 at diagnosis [31]. It has a relative 5-year survival of 70.3% independent of cancer stage. A total of 60% of VC cases are diagnosed as localised disease, with a 5-year relative survival of 86.2%, whereas 6% are diagnosed with distant metastases, with survival dropping to 22.9% [32]. In addition, VC exhibits a close association with HPV, with HPV 16, 13 and 18 showing incidences of 72.5%, 6.5% and 4.6%, respectively. [33]. In high-income countries, such as Denmark, Germany, Ireland, the Netherlands and the

United Kingdom, VC incidence has been rising among women of all ages, especially those under the age of 60 (a 5-yearly average increase of 11.6% from 1988 to 2007, $p = 0.02$) [34]. This trend is consistent with changing sexual habits and greater exposure to HPV infection in these particular age groups, and younger populations who were offered HPV vaccination are expected to have some level of protection against VC in the future [34].

The standard of care for treating early-stage gynaecological cancers is surgery, either with or without adjuvant chemotherapy and/or radiotherapy, depending on the operative margins and nodal status. Gynaecological cancers typically respond well to platinum-based combination chemotherapy (e.g., cisplatin and paclitaxel). Unfortunately, patients with disease that are resistant to or refractory to platinum-based therapies often have poor prognosis, with limited treatment options for advanced or recurrent gynaecological cancers after platinum-based chemotherapy has failed [13,31]. However, with the approval of new therapies, such as checkpoint inhibitors and targeted therapies in the metastatic disease setting, there is now an availability of new therapeutic agents which may have a positive impact on patient outcomes [35–37]. We will evaluate the current standard of care and treatment recommendations and consider recent developments in immunotherapy that could be utilised in the treatment of advanced disease in non-ovarian gynaecological cancers.

## 2. Methodology

To conduct this review, we reviewed the existing literature on the role of immunotherapy in non-ovarian gynaecological cancers. We specifically focused on endometrial cancer, cervical cancer and vulval cancer and explored recent trials and developments in immunotherapies. Literature searches were performed on Embase and MedLine from inception to 9 May 2023. In addition to screening relevant articles, we also searched through the references of included studies and reviews. This review consists of guidelines and recommendations from the Food and Drug Administration (FDA), the International Federation of Gynecology and Obstetrics, the British Gynaecological Cancer Society, the Royal College of Obstetricians and Gynaecologists, and the National Institute for Health and Care Excellence. Finally, we used the Boolean operators "AND/OR/NOT" and the following search terms (including synonyms and related words): "endometrial cancer", "cervical cancer", "vulval cancer", "immunotherapy" and "immune checkpoint inhibitors" to narrow down relevant articles.

## 3. Current Standard of Care

In vulval cancers, the International Federation of Gynecology and Obstetrics (FIGO) staging approach, which considers the lesion's size, depth of invasion (DOI) and involvement of the inguinal lymph nodes, is often used [16,38]. The FIGO staging has a major impact on both treatment and prognosis. For instance, the 5-year overall survival (OS) varies from 85.5% for FIGO I/II tumours to 20.3% for those with distant metastases. The FIGO staging is also standardised across all gynaecological cancers and accurately reflects the severity of disease progression, as evidenced by the difference in survival rates between different stages of disease [31].

The definitive treatment of EC in childbearing-age women is total hysterectomy and bilateral salpingo-oophorectomy, with or without lymphadenectomy [7,39]. Although this strategy is associated with a 93% 5-year survival rate, it causes a permanent loss of reproductive capacity in young females. Conservative treatment with oral progestins with medroxyprogesterone acetate (MPA) or megestrol acetate, though not a standard treatment, may be an alternative in women who strongly desire to preserve fertility after carefully examining the clinical and pathological characteristics of the tumour and having a multidisciplinary discussion [7,40], with a response rate of 75% but recurrence at 30–40% over a 4–357-month (median range) follow-up period [41,42].

In unresectable or surgically contraindicated cases, radical radiotherapy, such as intrauterine brachytherapy with or without external beam radiation therapy (EBRT), could be used as a palliative treatment, depending on tumour grade, and the two-year local control

rates are over 90% in stage I disease [43,44]. In advanced EC (FIGO stages III–IV), surgery in combination with adjuvant chemotherapy (with or without radiotherapy) is considered the gold standard of treatment. Surgical resection of pelvic and para-aortic lymph nodes should not be carried out unless the lymph nodes are deemed suspicious [39]. In unresectable, locally advanced disease without distant metastases, definitive radiotherapy (ERBT to the pelvis and image-guided brachytherapy) or neoadjuvant chemotherapy followed by surgery is advised. Concurrent chemotherapy may be beneficial in enhancing the effects of radiotherapy. Adjuvant chemotherapy may also be indicated following local treatment to prevent the development of distant metastasis [39].

When the disease becomes metastatic or recurrent, systemic treatment with hormone therapy (MPA, progestational agents such as megestrol acetate, selective oestrogen receptor modulators and luteinising hormone releasing antagonists) is preferred in low-grade carcinomas without rapidly progressive disease [39,45]. Some patients with hormone receptor-negative tumours may also derive a benefit from systemic hormone therapies. Carboplatin and paclitaxel are the main chemotherapy agents used for advanced recurrent EC. In locoregional recurrence, the primary treatment is EBRT with brachytherapy, with or without chemotherapy [39]. Palliative radiotherapy in the form of hypo-fractionated (smaller number of treatments with increased daily dose) small-volume EBRT can be performed in patients not fit for radical treatment [39]. Palliative surgery may also be beneficial in alleviating the symptoms of recurrent or metastatic disease. According to the most recent 2020 joint guidelines from the European Society of Gynaecological Oncology (ESGO), the European Society for Radiotherapy and Oncology (ESTRO), and the European Society of Pathology (ESP), pembrolizumab-based anti-PD-1 immunotherapy and the multi-tyrokinase inhibitor lenvatinib are now recognised as a second-line treatment for MSI/MMRd (MMR-deficient) metastatic or recurrent EC, but its use may be constrained due to regulatory approvals in various countries [39] (Table S1 [39,45,46]).

In CC, surgery and radical chemoradiotherapy are both potential primary treatment options (Table S2 [47–50]). The most common surgical treatments for CC include radical hysterectomy, total simple hysterectomy and cervical conization. Adjuvant radiation may be required if there are elevated recurrence risks, such as positive pelvic nodes, positive surgical margins and parametrial invasion [17]. Neoadjuvant chemotherapy may be used when radiotherapy facilities are unavailable to downstage the tumour and maximise radical curability, but it may obscure pathological findings and create false security [17]. For FIGO IB3 to IIA2 staged lesions, which are likely to require post-operative radiotherapy, concurrent platinum-based chemoradiation (CCRT) is preferred to avoid compounding treatment-related morbidity. Studies have shown that CCRT is more effective than radiotherapy alone, in terms of both progression-free survival (PFS) and overall survival (OS), and thus is recommended as the standard of care for locally advanced CC [44,51–56]. In metastatic or recurrent CC, palliative chemotherapy with paclitaxel and cisplatin is an active regimen with acceptable toxicity and high efficacy, with a 28% response rate and a median OS of 12.8 months [48,57]. The addition of bevacizumab, a recombinant humanised anti-vascular endothelial growth factor (VEGF) monoclonal antibody, demonstrated significant clinical benefits, extending the median OS by 3 months [57], and is now the preferred first-line regimen in metastatic or recurrent CC in combination with chemotherapy [48].

Surgery is commonly the recommended treatment for VC. Radiotherapy, with or without chemotherapy, may also considered as an alternative for advanced VC to avoid exenterative surgery [13,31]. However, because of the disease's inherent rarity and inter-study variations in inclusion criteria and the lack of prognostic markers, there is very little evidence for the most effective treatment at different stages [14,31] (Table S3 [58–60]).

To summarise, the treatment for EC, CC and VC varies; however, the main therapeutic interventions include surgery, platinum-based chemotherapy and radiotherapy. Additionally, there is a growing utilisation of checkpoint inhibitors in advanced disease, with anti-PD-1 inhibitors being particularly prominent as immunotherapeutic drug targets in the treatment of non-ovarian gynaecological cancers.

## 4. Genomic Alterations in Gynaecological Cancers

### 4.1. Endometrial Cancer

According to The Cancer Genome Atlas (TCGA [61]) Research Network, ECs can be classified into four groups according to their genetic and molecular information (Figure 1):

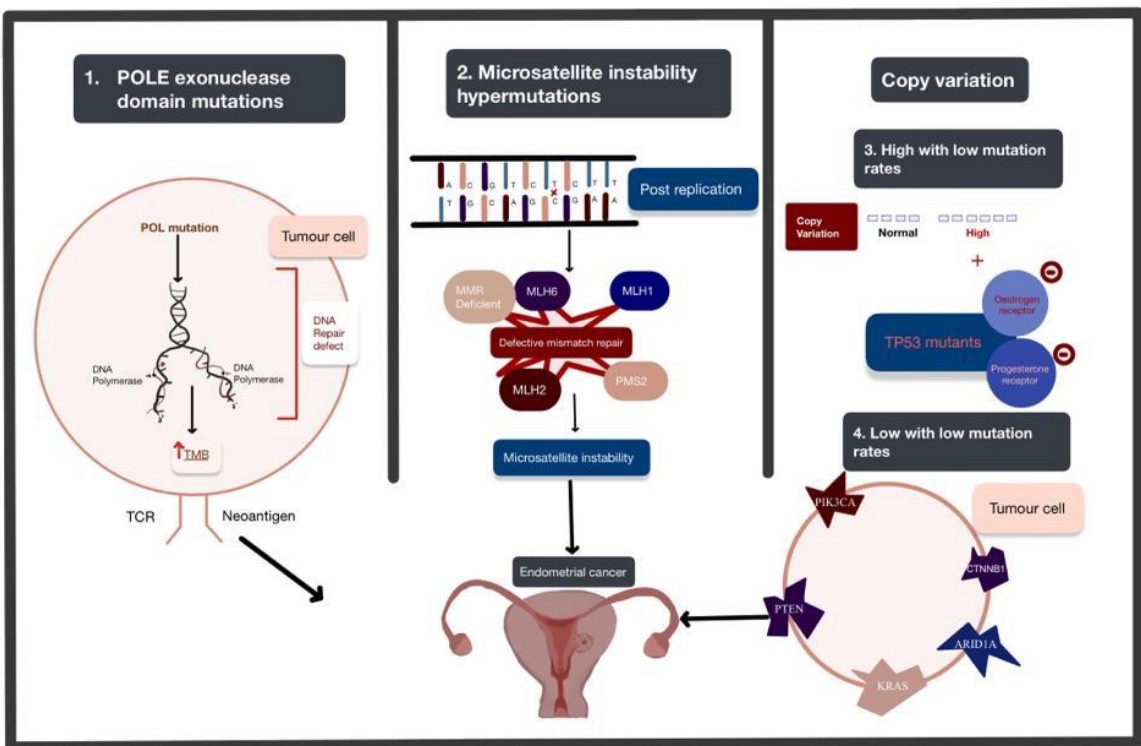

**Figure 1.** The four genomic alterations expressed in endometrial cancer (EC): POLE exonuclease domain mutations, microsatellite instability hypermutated, copy-number high with low mutation rate, and copy-number low with low mutation rate. The arrows used in this figure depict the order of the progression of these mutations from genetic alteration to the development of EC. Abbreviations: POLE: polymerase Epsilon; POL: polymerase; TMB: tumour mutational burden; TCR: T cell receptor.

Polymerase Epsilon (POLE) ultra-mutated: POLE encodes central catalytic and proof-reading subunits of DNA polymerase epsilon involved in leading-strand DNA replication. This subtype is characterised by POLE exonuclease domain mutations (EDMs) mostly in hotspot regions, more than 20% C > A transversions and unusually high mutation rates ($232 \times 10^{-6}$ mutations per Mb). The exonuclease proofreading function locates and replaces erroneous bases in the daughter strand through complementary pairing and ensures a low mutation rate. In endometrial carcinomas, mutations in DNA polymerases inactivate or suppress the proofreading abilities and increase replicative error rates [62]. It is identified in less than 10% [63] of ECs and is associated with aggressive histopathological features but high 5-year recurrence-free survival of around 90% [64].

Microsatellite instability hypermutated (MSI-H): MSI arises from alteration of the MLH1, MSH2, MSH6 and PMS2 genes involved in the post-replicative DNA mismatch repair system. MMR corrects DNA mismatches generated during replication, thereby preventing mutations from becoming permanent [65] in dividing cells, and its defects increase the spontaneous mutation rate [66]. MMR deficiencies can result from inheritance (as in Lynch syndrome), somatic mutations or epigenetic alteration.

The copy-number high with low mutation rate subgroup is characterised by significantly reoccurring amplifications or deletion regions, frequent mutation of TP53, and low oestrogen and progesterone receptor expression. This subgroup shares the same features as basal-like breast cancer and serous ovarian carcinoma.

The copy-number low with low mutation rate subgroup includes the rest of the tumours that do not belong to other groups and are frequently associated with mutations in PTEN, CTNNB1, PIK3CA, ARID1A and KRAS [67]. This subtype appears to share a pathogenesis with colorectal tumours [68].

### 4.2. Cervical Cancer

The Cancer Genome Atlas (TCGA) Research Network discovered novel amplifications [69] in immune targets CD274/PD-L1 and PDCD1LG2/PD-L2 and the BCAR4 lncRNA in CCs (Figure 2).

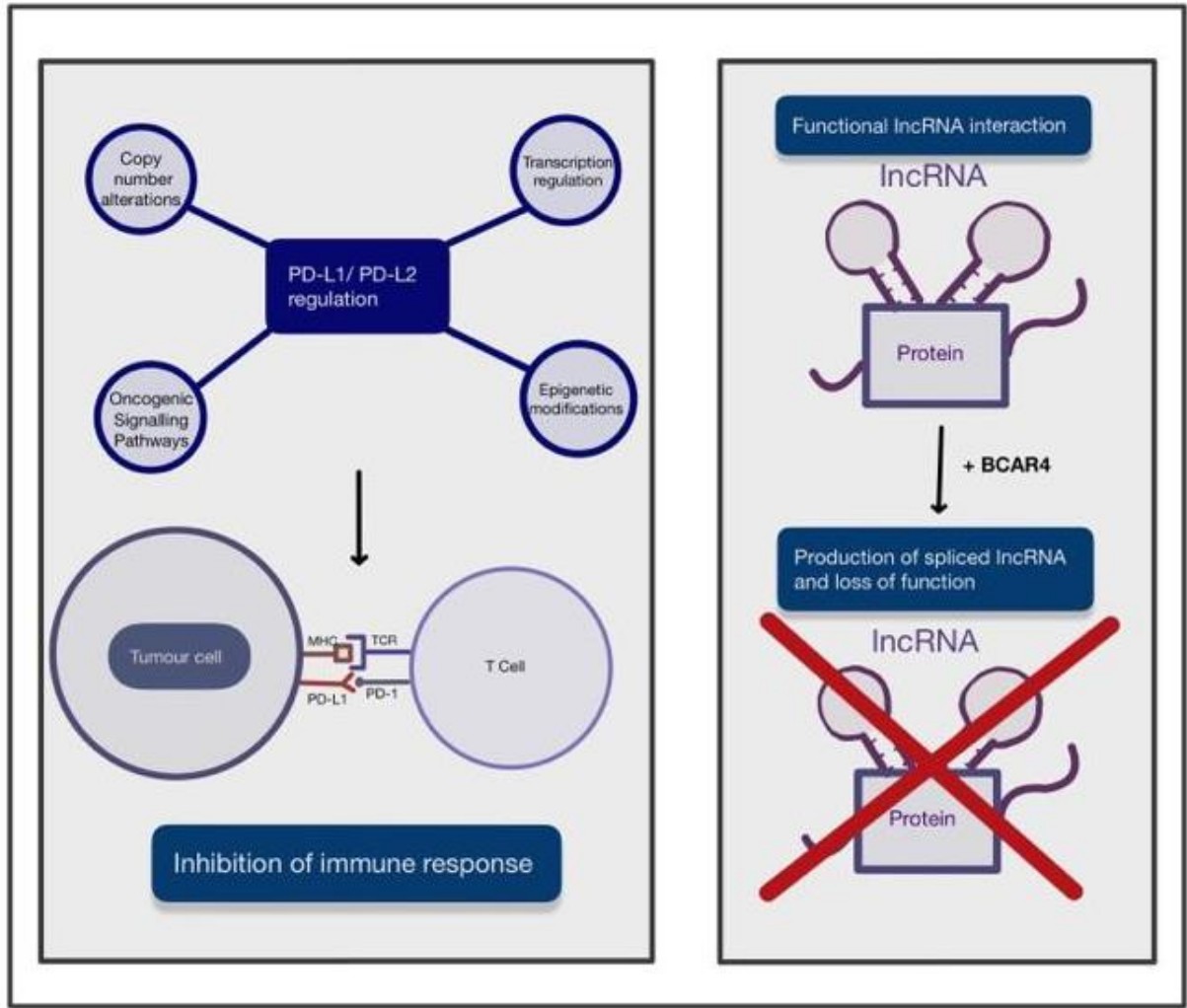

**Figure 2.** A figure depicting the two main novel amplications in the development of cervical cancer (CC). These include the upregulation of PD-L1/PD-L2 and BCAR4 expression and its role in disrupting the interaction of lncRNA with DNA, RNA and protein complexes. The arrows used in this figure depict the process of these novel amplifications and their effect on immune response or function. Abbreviations: PD-L1: programmed death ligand 1; PD-L2: programmed death ligand 2; MHC: major histocompatibility complex; TCR: tumour cell receptor; BRCA4: breast cancer anti-oestrogen resistance 4.

Programmed cell death ligand 1 (PD-L1) is a receptor mainly expressed on tumour cells and myeloid cells [70]. PD-L1 binds to receptor programmed cell death protein 1 (PD-1) on activated T cells and inhibits the immune response [71] towards tumour cells. PD-L2 is also a ligand of PD-1 and mainly present on antigen-presenting cells [72] and has an overlapping role with PD-L1. The regulation mechanisms of PD-L1/PD-L2 expression are complex and not yet fully understood. PD-L1 expression in tumour cells is regulated by various oncogenic signalling pathways and transcriptional and post-transcriptional factors [73]. Oncogenic factors that induce PD-L1 expression include the accumulation of oncogenic PIK3CA mutations and loss-of-function mutations of the negative regulator in the PI3K/Akt/mTOR pathway. In addition to these, there are also activation of the JAK/STAT pathway and the mitogen-activated protein kinase (MAPK) pathway by genetic mutations and growth factors [73]. Transcription factors, such as HIF-1, STAT3 and NF-κB, can bind to PD-L1 gene promoters, resulting in upregulation of its expression [73]. Finally, post-transcriptional regulation can be disrupted when microRNAs bind to mRNA, resulting in translational repression or alteration [73]. PD-L1 could play a significant role in the future management of CC, as HPV-positivity has been strongly correlated with an increase in PD-1 expression. This has been associated with an improved prognosis and a reduced chance of locoregional recurrence in HPV-positive cancers [74]. Additionally, a correlation has been found between high expression of PD-L1 and increased radiosensitivity, which may contribute to the improved prognostic rates seen in patients with high PD-L1 expression compared to patients with low PD-L1 expression [75]. Copy-number alterations at the 9p24.1 chromosomal region, transcriptional regulation by interferon factors, oncogenic signalling pathways, and epigenetic modifications by enzymes such as histone deacetylases may all contribute to regulation of PD-L1/PD-L2 [73]. In the case of CC, it has been shown that PD-L1 is significantly upregulated in productive HPV infection of the cervix.

Non-coding RNAs (ncRNAs) are RNAs that do not translate into a protein but are responsible for epigenetic regulation [76] at transcriptional and post-transcriptional levels, such as histone modification, DNA methylation and transcriptional silencing. NcRNAs can be categorised by size, and those longer than 200 nucleotides [77] are long ncRNAs (lncRNAs). LncRNAs could drive many important pathological cancer phenotypes [78] through their interactions with other molecules, such as DNA, protein and RNA. For instance, chromatin-bound lncRNAs can regulate gene expression; lncRNA interactions with multiple proteins can promote or impair the assembly of protein complexes; and lncRNAs can recruit proteins involved in mRNA metabolism. The breast cancer anti-oestrogen resistance 4 (BCAR4) gene was previously only studied in the context of breast cancer. BCAR4 produces a spliced lncRNA that is inversely associated with the development of resistance to anti-oestrogens in breast cancer cells and poor disease-free survival for recurrent breast cancer. Therefore, lncRNA BCAR4 was considered to play an important role in the metastasis and tamoxifen-resistance of breast cancer [79]. Over the years, more published evidence has suggested that the prognostic significance of BCAR4 also applies to gastrointestinal malignancy, breast cancer, osteosarcoma and various other cancer types [80]. It has been reported that BCAR4 expression was significantly upregulated in CC tissue and that patients with high BCAR4 expression showed worse survival outcomes. Moreover, overexpression of BCAR4 remarkably promoted the proliferation and motility of CC cells and the epithelial-mesenchymal transition process, while silencing BCAR4 had the reverse effect [81].

### 4.3. Vulval Cancer

VCs can develop via two major pathogenetic pathways [82,83], with 30–60% of vulvar tumours being related to HPV infection and the rest having no association with HPV. Both genetic mutations, such as TP53 mutation, and HPV infection can result in similar molecular effects [84], namely, inactivation of p53 and escape from apoptosis. Disruption of the TP53 tumour suppressor gene by point mutations plays a crucial role in VC, with the majority of mutations being single-point mutations in the TP53 core DNA-binding domain [85,86]. The p53 protein is essential for cellular responses under stress, such as DNA damage, hypoxia and oncogene activation. During cellular stresses, p53 binds to DNA as a tetramer by sequence matching and results in gene transcriptional regulation. This process leads to key cellular processes, such as DNA repair, cell-cycle arrest, senescence and apoptosis [87]. Thus, p53 is essential for tumour suppression and can be a potential target for cancer therapy [88].

Other than TP53 mutation as described in Figure 3, it has been reported that PD-L1 expression in squamous cell carcinoma of the vulva could be observed in 32.9% of patients, and its expression in peritumoural immune cells was confirmed in 91.4% of patients [89]. PD-L1 expression upregulation as described in Figure 3, has been suggested to be important for other genomic alterations or modifications described in vulval cancer.

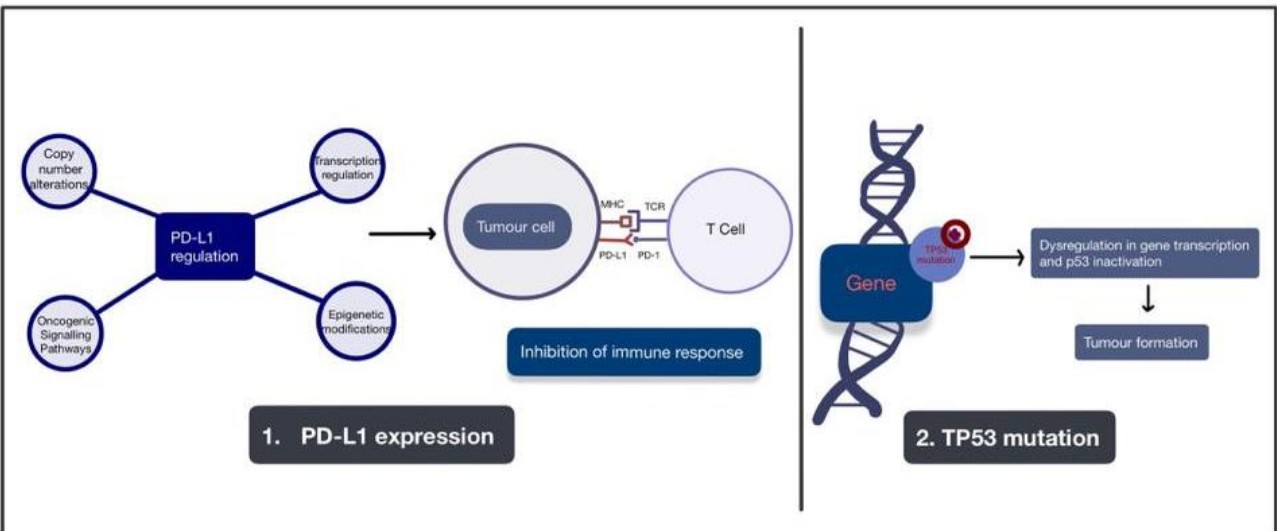

**Figure 3.** A figure depicting the two major pathogenic pathways involved with the development of vulval cancer. These pathways are the upregulation of PD-L1 expression and mutation of the TP53 gene. Arrows are used to depict the development of tumour formation as a result of these alterations. Abbreviations: PD-L1: programmed death ligand 1; PD-L2: programmed death ligand 2; MHC: major histocompatibility complex; TCR: tumour cell receptor; TP53: tumour protein 53.

Given the extensive knowledge of genomic alterations in EC, CC and VC, it is important that clinicians understand the molecular mechanisms underlying these cancers and use this knowledge to guide approaches to treatment whilst considering approaches to treatment, as well as how these alterations can affect disease progression and treatment efficacy.

## 5. Rationale for Immunotherapy in Gynaecological Cancers

Immune checkpoint inhibitors (ICIs) are a novel class of immunotherapy drugs that inhibit immune checkpoints, which, in turn, stimulate the immune system to recognise and attack cancer cells, thus eliciting an anti-tumour response [90]. This mechanism is utilised in the cases of PD-1 and PD-L1 inhibitors, where checkpoint inhibitors prevent the receptor binding of PD-1 to its ligand receptors PD-L1 and PD-L2, which are expressed on tumour cells, facilitating anti-tumoural activity. A few examples of approved drugs that target this specific checkpoint are pembrolizumab, nivolumab and retifanlimab, which was recently approved by the FDA, in March 2023 [91]. ICIs have demonstrated great promise in the management of a wide range of cancers, especially in advanced stages when treatment options become scarce, for example, metastatic melanoma [92], renal carcinomas [93] and non-small cell lung cancers [94,95], and thus there has been interest in potentially using this approach in managing gynaecological cancers [96]. Whilst immunotherapy is widely becoming an integral part of cancer treatment, only about 20–30% of patients achieve durable response [92]. Therefore, it is crucial to understand how certain alterations and a better understanding of the tumour microenvironment (TME) could impact response and overall survival in these patients.

### 5.1. Endometrial Cancer

The rationale for adopting immunotherapy in the management of EC is supported by the high immunogenicity of these tumours [97]. Although tumour mutational burden (TMB) is considered high in gynaecological cancers, the TMB in EC is observed to be higher than in cervical and ovarian malignancies [98]. This is significant, as a highly mutated tumour is thought to host more neoantigens that may be attractive to immune-based therapies [99]. Genes that have demonstrated high TMBs include DNA mismatch repair pathway genes (MSH2, MSH6, MLH1 and PMS2) and DNA polymerase epsilon (POLE) [99,100]. High MSI status has been observed in ECs [101], which has been associated with poor prognosis [102,103]. One phase II trial investigated ECs with MMR deficiency and concluded that MMR-deficient ECs were more responsive to PD-L1 blockade and would therefore benefit more from anti-PD-L1 therapy [104]. A study discovered that PD-L1 was expressed in 70 to 80% of EC samples, which is one of the highest levels among gynaecological cancers [105,106]. Blocking PD-L1 is an important mechanism of EC treatment, as it may be a favourable prognostic biomarker for immunotherapy response [106]. However, the use of PD-L1 inhibitors has not yielded striking results; therefore, further well-designed cohort studies are required to establish the efficacy of PD-L1 targeted therapy.

### 5.2. Cervical Cancer

The implementation of immunotherapy in CC is thought to be strongly supported by multiple molecular features, such as TMB, MSI, the presence of PD-L1/PD-1 and high tumour inflammatory state [107]. Up to 95% of CCs are thought to be associated with the oncogenic HPV infection, which is thought to drive carcinogenesis by hyperactivation of inflammatory pathways [108,109]. Chronic inflammation has been linked to the development of cancer, through the activation of pro-inflammatory mediators, such as cytokines and chemokines [110]. Moreover, the HPV-related E6 oncoprotein has been reported to express high levels of VEGF, a potent angiogenic factor in tumorigenesis [111]. Interleukin-6 (IL-6) is a cytokine which has been reported to increase the development of HPV-related cervical lesions and promote the expression of VEGF [112]. Bevacizumab is a human monoclonal antibody which inactivates the VEGF receptor, thus inhibiting endothelial cell proliferation and new vessel formation [113]. Studies have extensively found that bevacizumab significantly improved the mortality and prognosis of patients with advanced CC compared with chemotherapy alone [114–116]. Therefore, targeting the mechanisms of immune escape that target the HPV oncoproteins could be hugely promising in managing this disease.

CCs are known to have a high tumour mutational burden (TMB), which may be associated with a favourable response with ICIs [117]. A popular checkpoint that has received significant interest is PD-L1/PD-1, which is thought to be involved in HPV-related carcinogenesis by down-regulating T-cell activity [118]. One recent study reported that 57.4% of all cervical carcinomas investigated were PD-L1-positive [119], with another study reporting over-expression of PD-L1 in HPV-associated cancers [120].

### 5.3. Vulvar Cancer

Fewer studies have investigated the use of ICIs in VC, likely due to the rarity of the disease. The TME of VCs is poorly understood, but frequent expression of PD-L1 warrants the use of ICIs in managing the disease [121]. One study evaluated the expression of PD-L1 in VCs and found that 72.7% of tumours exhibited PD-L1, with 27.3% showing moderate-to-strong PD-L1 expression [122]. Another study reported that 44% of vulval squamous cell carcinomas investigated expressed PD-L1, which suggested a poorer prognostic indicator for OS [123]. Therefore, high PD-L1 expression in vulval carcinomas provides a strong rationale for the investigation of anti-PD-L1 therapies in future clinical trials [124]. Research has revealed that mutation in the tumour suppressor gene TP53 can be considered a prognostic biomarker for improved OS with ICIs in cancers, such as non-small cell lung cancer [125], renal cell carcinomas and melanomas [126]. Various studies concluded that TP53 plays a vital role in a large proportion of VCs by driving the process of tumorigenesis [127–129]. A recent meta-analysis explored the prognostic value of TP53 and reported that individuals with a TP53-positive tumour of the vulva had reduced OS compared with individuals with TP53-negative tumours [130]. Thus, there are promising results to suggest that TP53 may be a valuable prognostic indicator of response to possible future ICI therapies.

Immunotherapy is an evolving treatment modality in EC, CC and VC. Its ability to evade tumourgenesis and tumour proliferation through modulatory effects on key checkpoints has provided clinicians with alternative treatments that may optimise patients' outcomes, especially in advanced or metastatic disease.

## 6. Current Checkpoints in Clinical Practice

### 6.1. Endometrial Cancer

Pembrolizumab was first evaluated in 2015 by Le et al. in patients with metastatic disease, regardless of MMR status [104]. In a dMMR cohort, which included two EC cases, they reported initial overall response rate (ORR) and PFS values of 71% and 67%, respectively [104]. The Food and Drug Administration (FDA) later approved pembrolizumab for use in unresectable or metastatic dMMR and MSI-H EC, in 2017, based on five clinical trial outcomes in dMMR/MSI-H solid tumours (KEYNOTE-016, KEYNOTE-164, KEYNOTE-012, KEYNOTE-028 and KEYNOTE-158) [131–135]. The recommended dose is 200mg intravenously every 3 weeks or 400 mg every 6 weeks. From these trials, in PD-L1-positive EC (1/24 MSI-H status), the reported ORR and PFS were 13% (95% CI: 2.8% to 33.6%) and 1.8 months (95% CI: 1.6 to 2.7 months) [134]; in MSI-H/dMMR EC, they were 57.1% (95% CI: 42.2% to 71.2%) and 25.7 months (95% CI: 4.9 to not reached) [135].

Recent clinical trials have also explored pembrolizumab and lenvatinib combination therapies as well as standard chemotherapy carboplatin/paclitaxel. In patients with advanced EC, regardless of MMR status, the combination of pembrolizumab and lenvatinib significantly improved PFS and OS by 2.8 months in comparison to chemotherapy, along with a 15% higher ORR, according to the multicentre, open-label, randomised, phase III trial KEYNOTE-775/Study 309 (NCT03517449) with a population of 827 patients [8]. In 2021, the FDA fully approved pembrolizumab with lenvatinib as the new standard of care for patients with advanced EC who were previously treated with platinum-based chemotherapy based on the encouraging results [8]. Additionally, using pembrolizumab in combination with carboplatin and paclitaxel for advanced or recurrent EC demonstrated a significantly higher ORR 74.4% (32/43) than historic controls (*p* = 0.001), with a median PFS

of 10.6 months (95% CI: 8.3 to 13.9 months), according to a different phase II trial conducted by the Big Ten Cancer Research Consortium (BTCRC) (NCT02549209) [136].

Dostarlimab, a humanised anti-PD-1 IgG4 monoclonal antibody, demonstrated durable anti-tumour efficacy in the multicentre, phase I GARNET study (NCT02715284), with an ORR of 42.6% (95% CI: 30.6% to 54.6%) [137]. In a study population of 71 individuals with dMMR EC, complete therapeutic response was reported in 9 patients and partial response in 21 patients [137]. The promising results led to dostarlimab's accelerated approval by the FDA in April 2021 for use in recurrent or advanced dMMR EC. At the same time, the European Medicines Agency (EMA) also authorised conditional marketing for dostarlimab in April 2021, making it the first anti-PD-1 therapy for recurrent or advanced dMMR/HIS-H EC in Europe [138]. Another randomised, double-blinded, phase III trial, NSGO-RUBY (NCT03981796), is now ongoing to compare the efficacy and safety of dostarlimab plus carboplatin-paclitaxel chemotherapy in recurrent or primary advanced EC to chemotherapy alone [139].

Other immunotherapies, such as atezolizumab, avelumab and durvalumab, are anti-PD-L1 monoclonal antibodies. The FDA has already approved the use of these three monoclonal antibodies to treat metastatic or advanced urothelial carcinoma [140]. For 15, 15 and 35 patients with advanced/recurrent dMMR EC, atezolizumab, avelumab and durvalumab demonstrated ORRs of 13%, 26.7% (95% CI, 7.8% to 55.1%) and 40% (95% CI, 26% to 56%), respectively, suggesting durable clinical benefits in some patients. However, due to the small sample size, more research is necessary to confirm their clinical efficacies [141–143].

### 6.2. Cervical Cancer

In KEYNOTE-028 (NCT02054806), a multicentre, phase Ib, single-arm clinical trial, pembrolizumab (MK-3475) was given at 10 mg/kg every two weeks for up to 24 months to a cohort of 24 patients with PD-L1-positive advanced CC. The findings showed pembrolizumab's promising anti-tumour activity, with a 17% ORR (95% CI: 5% to 37%) and a median OS of 11 months (range: 1.3 to 32.2 months) at a tolerable safety profile consistent with that found in other tumour types [144]. Then, 98 patients with advanced CC who received previous treatments were enrolled in the multicohort, open-label, multinational phase II trial KEYNOTE-158 (NCT02628067), of which 82 were PD-L1 positive. The interim data analysis revealed that pembrolizumab had a 14.6% ORR (95% CI: 7.8% to 24.2%) in tumours that were PD-L1-positive and a 12.2% ORR overall (95% CI: 6.5% to 20.4%) in advanced CC patients who had previously received platinum-based therapy. Pembrolizumab was then granted accelerated approval by the FDA in June 2018 as a second-line treatment in PD-L1-positive recurrent or metastatic CC, making it the first sole immunotherapy approved for the treatment of advanced cervical malignancies [145]. Other studies have already begun to assess the possibility of using pembrolizumab in combination therapy, for example, the phase III clinical trial (NCT04221945) comparing pembrolizumab plus CCRT to placebo plus CCRT which is currently recruiting participants, with results being awaited in a few years [146], as well as an ongoing single-arm, phase II trial (NCT03444376) investigating the potential of the GX-188E HPV DNA vaccine plus pembrolizumab in recurrent or advanced CC to induce anti-tumour activity by provoking HPV E6-specific and E7-specific T cell responses [147].

Numerous studies have investigated the efficacy of other ICIs in CC. In the phase II NRG-GY002 study (NCT02257528), nivolumab, an anti-PD-1 antibody authorised by the FDA in 2014 for use in melanoma, was evaluated for persistent or recurrent CC. Results from 26 patients showed poor anti-tumour efficacy and an acceptable safety profile [148]. The CheckMate-358 trial (NCT02488759) showed promising efficacy of nivolumab in patients with recurrent or metastatic cervical, vaginal or vulvar carcinomas, with a median OS of 21.9 months (95% CI: 15.1 months to not reached) in the CC cohort and ORRs of 26.3% (95% CI: 9.1 to 51.2) for CC and 20.0% for VC (95% CI: 0.5 to 71.6) [149]. However, the sample size was small ($n = 24$; cervical = 19; vaginal/vulvar = 5) and more justifications are needed. Cemiplimab, another anti-PD-1 monoclonal antibody approved to treat squa-

mous cell carcinoma in lung and skin cancers [150], was also shown to have preliminary clinical activity in recurrent CC. Cemiplimab and chemotherapy were compared in the international multicentre, randomised, phase III trial GOG-3016 (NCT03257267), which included 608 patients (304 in each arm) and ran between 2017 and 2020. The cemiplimab group outperformed the chemotherapy group significantly in terms of ORR (16.4%, 95% CI: 12.5 to 21.1 versus 6.3%, 95% CI: 3.8 to 9.6) and OS (12 months versus 8.5 months; hazard ratio = 0.69, 95% CI: 0.56 to 0.84, two-sided $p < 0.001$) [151]. This study was the largest randomised, controlled, phase III clinical trial to assess cemiplimab as a second-line treatment for recurrent or metastatic CC and significantly proved the its clinical benefit over chemotherapy.

Anti-PD-L1 agents, such as atezolizumab, avelumab and durvalumab, are currently in clinical trials with relatively limited clinical evidence for their efficacy in CC to date. Several phase II clinical trials (NCT03340376, NCT03614949 and NCT02921269) have been conducted to evaluate the efficacy of atezolizumab in combination therapies with chemotherapy, radiotherapy and bevacizumab [152]. Similarly, avelumab is being studied in ongoing clinical trials (NCT03260023 and NCT03217747) for its usefulness in combination with the TG4001 HPV vaccine [153] or chemoradiation [154]. In addition, durvalumab in combination with ADXS11-001 (NCT02291055) [155], stereotactic ablative radiation and tremelimumab (NCT03452332) are currently being explored in both cervical and vulval malignancies [152].

Monoclonal antibodies, such as the anti-VEGF antibody bevacizumab, are also crucial in the suppression of tumour angiogenesis, such as the anti-VEGF antibody bevacizumab. Based on interim analysis from the controlled, randomised, phase III clinical trial GOG 240 (NCT00803062), bevacizumab was approved by the FDA and the United Kingdom's Cancer Drug Fund in 2014 to be used in combination with chemotherapy for women with recurrent or metastatic CC [156]. The addition of bevacizumab significantly increased median OS from 13.3 months to 16.8 months in a study involving 452 patients ($p = 0.0068$) [156], which led to the 2017 ESMO guidelines and National Comprehensive Cancer Network (NCCN) Cervical Cancer Treatment Guidelines recommending paclitaxel-cisplatin with bevacizumab as the preferred first-line treatment for advanced CC [48,157].

In addition to PD-1/PD-L1, the CTLA-4 immune checkpoint is also critical in the response to T-cell activation. ICIs targeting CTLA-4 have been a major focus of immunotherapy development in several malignancies, including CC, with ipilimumab being the most widely researched anti-CTLA-4 monoclonal antibody for CC [158]. In the clinical trials GOG-9929 (NCT01711515) [159] and 938TiPA (NCT01693783) [160], ipilimumab demonstrated immune-modulating activity in significant T-cell population expansion as an adjuvant therapy to definitive CRT in women with locally advanced CC [159] but no significant response as a monotherapy in metastatic CC [160]. However, the GOG-9929 trial's study population was limited ($n = 21$), and additional research is required to confirm the therapeutic potential of anti-CTLA-4 therapy.

Furthermore, TIGIT, a protein expressed on both T and NK cells that modulates their activity and maturation is another crucial immunotherapy target [161]. TIGIT has a strong affinity for CD155, and, once attached, it suppresses the immune system, resulting in a significant increase in TIGIT+ cells in CC patients [162]. Preclinical studies have already shown the synergistic impact of anti-TIGIT with anti-PD-1/PD-L1 antibodies as a potential therapeutic method to treat CC [162]. Based on this, an ongoing phase II clinical trial (NCT04693234) is examining the anti-tumour activity and safety of the combined use of the anti-TIGIT antibody ociperlimab with the anti-PD-1 antibody tislelizumab in previously treated recurrent or metastatic CC [163].

### 6.3. Vulvar Cancer

Due to the rarity of the disease, clinical evidence of ICIs in VC is relatively limited. The KEYNOTE-028 (NCT02054806) study evaluated pembrolizumab in 20 different PD-L1-positive advanced solid cancers, including 18 vulvar squamous cell carcinomas (VSCCs) [164]. The results in VSCC patients were not as encouraging as in EC and CC; the median OS was 3.8 months (95% CI: 2.8 to 5.5 months), with an ORR less than 10% and a median PFS less than 5%. Only one patient had a partial response and the 6- and 12-month OS rates in the VC cohort were 42% and 28%, respectively [164]. Furthermore, five vulvar and four vaginal cases were enrolled in the phase I/II CheckMate-358 trial (NCT02488759) of nivolumab, and the results showed an ORR of 20% (95% CI: 0.5 to 71.6), with one patient showing partial response for five months [149]. Although nivolumab revealed a tolerable safety profile and statistically encouraging efficacy in recurrent or metastatic VC, the study sample size was too small and more investigation is necessary. Pembrolizumab with cisplatin-sensitised radiation therapy has been proposed as a combination therapy for unresectable, locally progressed or metastatic VC in an ongoing single-arm, phase II trial (NCT04430699) [165].

Taken together, we have established the key drug targets and immunotherapy agents currently in use in clinical practice (Table 1, Figure 4). These targeted therapies are crucial in regulating the tumour microenvironment and restoring immune response against tumour cells. Although most of these inhibitors are approved for use in EC and CC, it is likely there will be further advances in near future. Future developments could include novel checkpoint combination and targeted therapies as our understanding improves. This could result in more definite and effective therapies that will advance the management of VC.

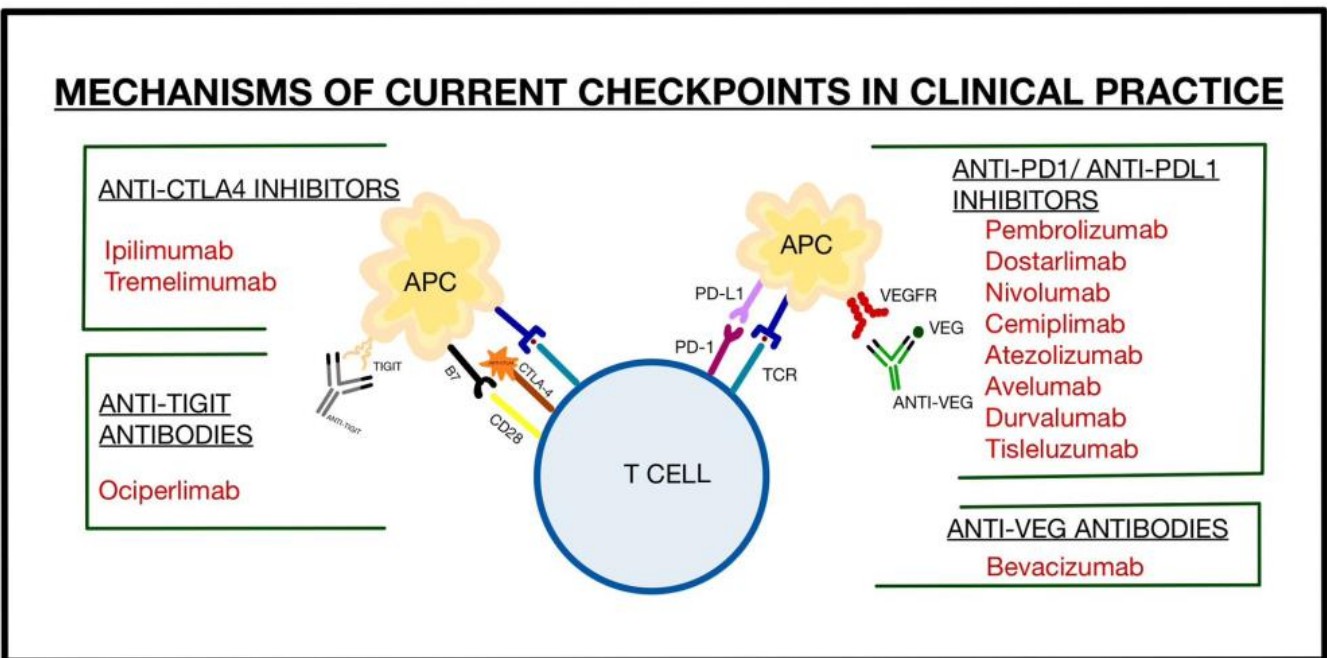

**Figure 4.** A figure summarising the main drug targets in EC, CC and VC and examples of the class of drugs that regulate these immune responses. Abbreviation: APC: antigen presenting cell.

**Table 1.** Summary of the immunotherapy agents in use for metastatic, unresectable or advanced EC, CC and VC.

| Cancer Type | Immunotherapy Agent | Treatment Recommendations |
|---|---|---|
| Endometrial Cancer | Pembrolizumab | Pembrolizumab is approved for use as a second-line treatment in unresectable or metastatic MSI/MMRd EC. It has also been shown to have an increased clinical benefit as a combination therapy over standard chemotherapy treatment (carboplatin/paclitaxel) when used as a second-line systemic treatment for microsatellite-stable carcinomas [8,36]. |
| | Dostarlimab | Dostarlimab is an anti-PD-1 inhibitor approved by the FDA as another second-line immunotherapy agent for patients with advanced or recurrent dMMR- MSI-H EC following its significant response rates in the phase I GARNET study [137]. As a combination therapy with carboplatin/paclitaxel, evidence has been generated of a decreased rate of disease-free progression compared to chemotherapy alone in patients with recurrent or advanced disease [139]. |
| Cervical Cancer | Pembrolizumab | Pembrolizumab is a second-line treatment in PD-L1-positive recurrent or metastatic CC. In clinical trials, it has shown a greater overall response rate when used in patients with PD-L1-positive tumours compared to platinum-based therapy [145]. Its efficacy as a combination therapy is currently being studied in trials [146,147]. |
| | Ipilimumab | Although ipilimumab has not shown significant response as a monotherapy in metastatic disease, durable responses have been observed when it has been used as an adjuvant therapy with definitive CRT in locally advanced CC [159]. |
| Vulvar Cancer | Pembrolizumab | Pembrolizumab with cisplatin-sensitised radiation therapy is currently being evaluated as a combination therapy for unresectable, local or metastatic VC in a phase II trial (NCT04430699) [165]. |
| | Nivolumab | Although nivolumab has been shown to have a notable efficacy and a tolerable safety profile in recurrent or metastatic VC, the study sample size was small [149]. Given the lack of sufficient evidence of the efficacy of ICIs in VC, clinicians should exercise clinical judgment when considering immunotherapy agents in the treatment of recurrent or metastatic VC. |

## 7. Checkpoints in Development (Human Trials) and in Pre-clinical Development (Animal Studies)

### 7.1. Adoptive T-Cell Therapy

Adoptive T cell therapy utilises genetically modified lymphocytes to target specific tumour markers [166]. Due to its increased specificity, it can be effective in preventing tumour proliferation and maintaining tumour regression. Interest in the use of adoptive T-cell therapy in advanced HPV-associated cancers, such as CC and VC, has grown, and recently the oncoproteins E6 and E7 have been recognised as potential targets for adoptive T-cell therapy in the treatment of cervical cancer [167–169]. HPV-16-positive tumour reactivity to E6 specific therapies was noted in metastatic anal cancer [170], leading to its application in the treatment of cervical cancer. The clinical benefits of adoptive T cell therapy have also been found in the clinical trial NCT01585428, where out of nine patients with metastatic cervical cancer, two experienced complete responses of 15 and 22 months, respectively. Another patient also experienced a partial response for 3 months, and the study found that increased frequencies of HPV-reactive T cells correlated with an improved clinical response [171]. A recent phase II trial demonstrated the feasibility of its use in the clinical setting in patients with cervical cancer, with 28% of cervical cancer patients experiencing an objective tumour response [172].

### 7.2. Cancer Vaccines

The efficacy of cancer vaccines in preventing cancer recurrence is currently being explored. A study in mice found that GDE7 mRNA-based vaccines were effective in invoking CD8+ T cell responses and controlling tumour progression, even in advanced cervical disease [173]. This finding may change the trajectory of cancer vaccines aimed at HPV-16-associated cancers, and with future studies, its efficacy in other strains of HPV may also prove to be productive.

Human studies also found that ISA101, a peptide vaccine developed for HPV-associated cancers, was successful in invoking promising response rates in patients when used in combination with nivolumab. The median duration of response was 11.2 months, with 3 out of 8 patients showing an objective response without progress at 3 years [174]. A phase I/IIa trial investigated the use of a folate-binding peptide vaccine and its efficacy in preventing cancer recurrence in ovarian and endometrial cancer [175]. It was found to stimulate the production of tumour-specific cytotoxic T cells and memory T cells. Although folate-binding proteins are rarely expressed in ovarian and endometrial tumour cells, it was found that an increased expression of this protein correlated with more aggressive disease [175]. The modalities of cancer vaccines are broad; however, the differences in their functionality may open doors in the management of treatment-resistant patients or those with recurrent disease [176–178].

### 7.3. Novel Immune-Based Immunotherapy

Current novel and emerging immunotherapy targets, such as lymphocyte activation gene-3- (LAG-3) and Galectin-3 (GAL-3), have not been extensively studied in non-ovarian gynaecological malignancies. LAG-3 is believed to be expressed in approximately 30% of patients with EC, demonstrating its potential as an attractive target for future therapies [179]. Following findings from RELATIVITY-047, relatlimab became the first FDA-approved LAG-3 inhibitor as a combination therapy with nivolumab for melanoma [179]. In endometrial cancer, LAG-3+ lymphocytes and GAL-3 neoplastic cells have been found to have an increased expression in mismatch repair-deficient tumours and have also shown a positive correlation in expression in nonmethylated mismatch repair-deficient endometrial carcinoma [180]. Recent studies have demonstrated that LAG-3 had an improved tumour response in patients when delivered as a dual therapy [181,182]. Dual therapies with PD-1 and LAG-3 have already been shown to be more effective in mediating tumour regression in other cancers and could be used alongside current immunotherapy inhibitors such as pembrolizumab to enhance anti-tumoural response in EC [183,184]. Novel targets such as

PI3Kδ have also been shown to facilitate tumour cell proliferation when used in combination with an anti-LAG3 antibody therapy [185]. An increased CD8:Treg ratio was required to promote tumour responsiveness to PI3Kδ inhibition and anti-LAG3 dual therapy; hence, the selective nature of therapeutic response may be beneficial in specific patients who express the required tumour molecular classification [185].

Other potential targets for endometrial cancer include T-cell immunoglobin-3 (tim-3) [179]. It is thought to have a similar cellular mechanism to PD-1 and decreases tumour mutational burden by disrupting cellular trogocytosis of CD8+ TILs [186]. PODIUM 204 is an ongoing phase II study which will compare the efficacy and safety of retifanlimab (anti-PD-1 therapy) as a monotherapy and as a dual therapy (in combination with other targets, such as LAG-3 and TIM-3) in advanced or metastatic EC. Findings from this study could promote the development of future studies on TIM-3 monotherapy and its safety and efficacy in the treatment of EC and other non-gynaecological malignances.

Finally, bispecific antibodies could transform the treatment of non-ovarian gynaecological carcinomas. NCT05032040, a phase II trial in advanced gynaecological cancers, is currently evaluating a bispecific monoclonal antibody that targets PD-1 and cytotoxic T-lymphocyte antigen-4 [187]. If shown to be effective, it could facilitate the development of other novel CTLA-4 inhibitors, such as AGEN1181 [179]. AGEN1181 has already demonstrated clinical activity as a monotherapy and in combination with balstilimab (anti-PD-1) in patients with pre-treated microsatellite-stable metastatic colorectal cancer [179,188].

### 7.4. Intra-Tumoural Oncolytic Viral Therapy

Oncolytic viral therapy is another novel strategy for the management of non-ovarian gynaecological malignances. The mechanism of intra-tumoural oncolytic viral therapy is the injection of immunostimulatory agents that cause tumour cell lysis and evoke an immune response [189]. An in vitro study on the efficacy of oncolytic herpes simplex virus in cervical cancer resulted in a reduction in tumour growth and an increase in the number of CD8+ T cells [190]. Despite the promising result, the findings were not clinically significant [190]. The lack of development of oncolytic viral therapies for the treatment of EC, CC and VC makes it less likely that these therapies will be adopted as quickly as other novel therapies currently in development.

These various checkpoints in development for EC, CC and VC aim to enhance treatment outcomes and refine disease management and are summarised in Table 2.

**Table 2.** A summary of novel targets in development in clinical and pre-clinical studies on endometrial cancer, cervical cancer and vulval cancer.

| Therapy | | Summary of Novel Treatments |
|---------|---|------------------------------|
| Adoptive T cell therapy | E6 and E7 oncoprotein-based therapies | HPV-positive tumours have been evaluated as potential targets for E6 and E7 oncoprotein-based T cell therapy in CC and VC. Recent studies have shown promising results for their use in cervical cancer, which may elicit further interest in their development [171,172]. |
| Cancer vaccines | GDE7 mRNA-based vaccines | Preclinical studies have found that GDE7 mRNA-based vaccines may be effective in HPV-16-associated cancers, such as CC and VC, in controlling tumour proliferation and the effectiveness of CD8+ tumours [173]. |
| | ISA101 | ISA101 is a novel peptide protein developed for HPV-associated cancers. It has been successful when used in combination with nivolumab to prevent cancer recurrence, with a notable response occurring in EC despite the lack of expression of folate binding proteins in EC. |

**Table 2.** *Cont.*

| Therapy | | Summary of Novel Treatments |
|---|---|---|
| Novel immune-based immunotherapy | LAG-3 | LAG-3 has been found to be expressed mainly in mismatch repair-deficient tumours and in nonmethylated mismatch repair-deficient EC [180]. Dual therapies with PD-1 and LAG-3 have already been shown to be more effective in mediating tumour regression in other cancers and enhancing the anti-tumoural response [183,184]. |
| | GAL-3 | GAL-3 has also been shown to have an increased expression in mismatch repair-deficient tumours and nonmethylated mismatch repair-deficient EC [180], and it is likely that specific therapies will be developed utilising its potential as a novel target. |
| | PI3Kδ | PI3Kδ can facilitate tumour cell proliferation when used in combination with an anti-LAG3 antibody therapy; however, an increased CD8:Treg ratio is required to see an increased response from this dual therapy in non-ovarian gynaecological cancers [185]. |
| | TIM-3 | TIM-3 can decrease tumour mutational burden by disrupting cellular trogocytosis of CD8+ TILs, and an ongoing phase II study is comparing the efficacy and safety of retifanlimab (anti-PD-1 therapy) when used as a monotherapy or in combination with other novel targets in advanced EC. |
| | Bispecific monoclonal antibodies | Bispecific antibodies are currently being investigated in a phase II trial on advanced gynaecological cancers. If found to have sufficient efficacy, they could be used to exploit dual targets, such as PD-1 and CTLA-4 [187]. |
| Intra-tumoural oncolytic viral therapy | Oncolytic herpes simplex virus | Oncolytic herpes simplex viral therapy can stimulate a reduction in tumour growth and increase the number of CD8+ T cells in CC [189]. However, the statistical significance of the results ($p = 0.03$) is insufficient to attract significant interest in it as an alternative immunotherapy agent to existing treatments [189]. |

## 8. Future Directions

Although there have been advances in the management of EC, CC and VC, there is still progress to be made, especially in patients with advanced disease. Novel, modern neoantigen targeted immunotherapies need to be identified to increase the efficacy of immunotherapies. An emerging target, TIM-3, has been shown to be expressed in EC and CC [191,192]. Tim-3 exerts immunosuppressive effects by increasing regulatory T cell activity and promoting cytotoxic T cell exhaustion [191]. In EC, tumoural expression was mainly found in mismatch repair groups, with prominent expression in patients with MLH1-hypermethylated and MSH6-deficient groups. More studies on its response as a potential biomarker in non-ovarian gynaecological cancers are required, as our current understanding of TIM-3 is limited. TIM-3 has shown clinical benefit in cancers, such as melanoma and non-small cell lung cancer, and could be developed as an alternative immunotherapy for specific candidates with non-ovarian gynaecological cancers who do not respond well to existing immunotherapy treatment [186].

Other interesting therapies that should be explored further include the use of biomarkers such as IGF2R in targeted therapy. IGF2R is thought to have oncogenic properties in cervical cancer, and its upregulation in tumour cells has been linked to poorer prognosis [193]. Despite its apparent potential as a clinical target for therapeutic strategies, it has rarely

been explored as a targeted therapy in treatment. Future studies into its mechanism may be beneficial for patients with cervical cancer who express high levels of IGF2R [193].

The combination of chemotherapy with ICIs is becoming more common, and there are currently ongoing trials exploring the efficacy of these combination therapies [194]. Further directions for study include the use of concurrent chemoradiation alongside ICIs, which could prove more effective than monotherapy.

In cervical cancer there have been few phase III trials demonstrating the benefit of novel therapies over mainstay therapies such as chemotherapy [195,196]. An increase in phase III trials will be useful in giving clinicians more clarity and confidence when straying away from standardised treatment algorithms. It will also ensure that patients are more optimally managed without unnecessarily increasing their toxicity profiles.

Finally, due to the rare nature of vulval cancers, their TME is rarely understood. Pembrolizumab is often indicated due to the frequent expression of PD-L1 in vulval cancer [197,198]. The identification of patients who would truly benefit from immunotherapy will provide greater precision in the disease management of VC. Additionally, greater utilisation of other expressed biomarkers, such as TP53, in future therapies could provide a greater benefit than the current treatment algorithm for the management of VC.

## 9. Conclusions

While there has been a significant number of reviews exploring the advances in care of patients with gynaecological cancers, much of the focus has been on ovarian cancers. In this review, we have described state-of-the-art treatments for non-ovarian gynaecological cancers. We have examined the role of immunotherapy across the various disease states. Finally, we explored the direction of future advances in the treatment of these debilitating conditions affecting women around the world.

**Supplementary Materials:** The following supporting information can be downloaded at: https://www.mdpi.com/article/10.3390/futurepharmacol3020031/s1, Table S1: Summary of the treatment of endometrial cancer based on current guidelines [39,45,46]; Table S2: Summary of the treatment for cervical cancer based on tumour stages [47–50]; Table S3: Summary of the treatment recommendations for vulval cancer based on tumour stages and types [58–60]. References [39,45–50,58–60] are cited in the Supplementary Materials.

**Author Contributions:** Conceptualization, S.A. and C.M.; literature review, Y.G., S.C. and H.D.; validation, S.A., J.R.G. and C.M.; formal analysis, Y.G., S.O., S.C. and H.D.; resources, Y.G., S.O., S.C. and H.D.; data curation, Y.G., S.O., S.C. and H.D.; writing—original draft preparation, Y.G., S.O., S.C. and H.D.; writing—review and editing, S.A., Y.G., J.R.G. and C.M.; visualization, Y.G. and S.O.; supervision, S.A. and C.M. All authors have read and agreed to the published version of the manuscript.

**Funding:** This research received no external funding.

**Institutional Review Board Statement:** Not applicable.

**Informed Consent Statement:** Not applicable.

**Data Availability Statement:** The data presented in this study are available in manuscript and Supplementary Materials.

**Conflicts of Interest:** The authors declare no conflict of interest.

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
