# Peer review of "Current Methods and Advances in the Immunotherapy Treatment of Non-Ovarian Gynaecological Cancers"

_futurepharmacol, doi:10.3390/futurepharmacol3020031_

Round 1

Reviewer 1 Report

Important review of immunotherapy treatment options for non-ovarian gynaecological cancer.

1. Introduction
This section very lengthily summarises aetiology and incidence of endometrial, cervical and vulva cancers. There is no introduction to what immunotherapy is.  Deficit of knowledge regarding immunotherapy in EC, CC and VC should be commented on here, as well as what is known in gynae cancers. 

2. Current standard of care

First paragraph unclear that it is discussing vulva cancer until last line.

Second paragraph appears to discuss all gynae cancers. This would be better placed within Introduction section and expanded upon.

The remaining paragraphs appears to be about EC but not CC or VC.

Line 123: 'recurrence at 30-40%' over what time period follow-up?

Tables 1-3 are a well-presented treatment aid but too lengthy to include as tables within the manuscript. Consideration should be given to adding them as supplementary appendices.  If they were to be incorporated note these comments:

Table 1: Surgery part - no mention of debulking surgery which is a new standard of care for advanced stage disease, thereby making extra-uterine mets not a contraindication (as table states it is)

Table 2:

FIGO Stage 1A1: option of 2 loops not mentioned

FIGO Stage 1A2-1B2 section does not mention management of small volume 1B1 with conservative surgery (2 loops, simple TLH or simple trachelectomy)

Table 3: Vulva Paget's disease is pre-cancerous and therefore does not need to be included.

3. Genomic alterations section

no changes

4. Rationale for immunotherapy section

Explanation of what ICIs are needs to be given either here or in introduction.

Line 314: acronym 'TMB' used for the first time but it's explanation not given until line 331

5. Current Checkpoints section

Title of section is repeated twice

Line 373 'MMRd' is used but throughout rest of manuscript it is dMMR. Please standardise.

A table of ICIs in EC, CC amd VC would be very helpful here. 

6. Checkpoints in development section

no change

7. Future directions

no change

8. Conclusion

no change

Overall, the manuscript reads very 'top heavy' with lots of information about the cancers and current non-immunotherapy treatment and less on the immunotherapy treatment. Requires major revision with rewrite as above, explaining the field of immunotherapy and limitations of paucity of evidence in non-ovarian cancers etc. I note that no methodology (of how literature was sought for the review) is presented.

Reviewer 2 Report

In the manuscript: “Current methods and advances in the immunotherapy treatment of non-ovarian gynaecological cancers” The authors examine the current management of endometrial, cervical and vulval cancer and evaluate the novel therapies such as adoptive T-cell therapy, intra-tumoral oncolytic viral therapy and cancer vaccines which are under development. An interesting topic that contributes to knowledge in the area, but certain issues must be corrected.

Major revisions

1.    Please briefly describe the regulation mechanisms of PD-L1/PD-L2 expression (line 246)

2.    The authors should develop the theme further. For example, in line 250, the authors do not conclude the idea that PD-L1 is significantly upregulated in productive HPV infection.

3.    The authors must conclude each section by mentioning the importance of what is mentioned with respect to the topic under development. For instance, “Taken together this information….”

4.    Please, you must reference lines 324-326, where was this information taken from?

5.    Authors are encouraged to complete their figures with the therapies proposed in their manuscript so that the figures complement this information and not just present the main mechanisms by which non-ovarian gynecologic cancers develop.

6.    Authors should add a table listing new treatments reviewed by the authors in the manuscript for non-ovarian gynecologic cancers.

Minor revisions

7.    Authors must check the grammar of the manuscript. There are repeated words (line 39, 410).

In the manuscript: “Current methods and advances in the immunotherapy treatment of non-ovarian gynaecological cancers” The authors examine the current management of endometrial, cervical and vulval cancer and evaluate the novel therapies such as adoptive T-cell therapy, intra-tumoral oncolytic viral therapy and cancer vaccines which are under development. An interesting topic that contributes to knowledge in the area, but certain issues must be corrected.

Major revisions

1.    Please briefly describe the regulation mechanisms of PD-L1/PD-L2 expression (line 246)

2.    The authors should develop the theme further. For example, in line 250, the authors do not conclude the idea that PD-L1 is significantly upregulated in productive HPV infection.

3.    The authors must conclude each section by mentioning the importance of what is mentioned with respect to the topic under development. For instance, “Taken together this information….”

4.    Please, you must reference lines 324-326, where was this information taken from?

5.    Authors are encouraged to complete their figures with the therapies proposed in their manuscript so that the figures complement this information and not just present the main mechanisms by which non-ovarian gynecologic cancers develop.

6.    Authors should add a table listing new treatments reviewed by the authors in the manuscript for non-ovarian gynecologic cancers.

Minor revisions

7.    Authors must check the grammar of the manuscript. There are repeated words (line 39, 410).

Round 2

Reviewer 1 Report

Well done to the authors for taking on board all previous comments.  The manuscript reads much more succinctly. 

However, please can they address the following:

Lines 138-142:These 3 sentences all relate to vulva cancer and so that must be introduced in the first sentence.  The FOGO staging which considers inguinal node involvement pertains to vulva cancer and not 'gynaecological cancers' as stated in line 140

Line 167: EBRT can be used as 'palliative' not 'primary' treatment. Primary treatment suggests adjuvant treatment to follow

Line 206: In CC surgery and 'radical chemoradiotherapy' are both potential primary...Please change 'radiotherapy' to this, and delete 'either with or without adjuvant chemotherapy'

Lines 213-215: Definitive concurrent platinum-based chemoradiation (CCRT) is preferred. There is no role for definitive radiotherapy.

Table S1: Fisrt column 'definitive radiotherapy' should read 'palliative radiotherapy'

Second column under ovarian preservation comments on 'consider in pre-menopausal patient under 45 years'.  In practice this is under 40years

Table S2: FIGO1B3/IIA2 chemo and surgery/RT, neoadjuvant chemotherapy followed by surgery or RT should fall under Generally recommended (B) not strongly recommended (A)

Author Response

Dear reviewer, we have made changes per your suggestions highlighted in yellow in the uploaded manuscript. The changes in supplementary tables have also been corrected. 

Thank you. 
